# OpenReview forum: "Unbiased Reward Modeling from Implicit Feedback for LLM Alignment"
_ICML.cc/2026/Conference — ICML 2026 regular_

### Official Review · Reviewer_upri · 2026-02-16

**Soundness:** 3
**Presentation:** 4
**Significance:** 2
**Originality:** 3
**Overall Recommendation:** 4
**Confidence:** 4

**Summary:**

The authors aim to discuss the issue of learning reward models from implicit feedback (e.g., clicks, copy actions) instead of explicit preference labels. They formalize implicit reward modeling as a latent-variable problem where observed feedback is the product of true preference and user action, and propose ImplicitRM, which stratifies samples into four latent groups (positive/negative × active/passive) and optimizes an ELBO-derived objective.

**Compliance With Llm Reviewing Policy:**

Affirmed.

**Final Justification:**

The authors response have addressed my concerns. That said, I believe my current rating is adequate given the scope of their contributions.

**Key Questions For Authors:**

How is the degree of heterogeneity in positive actions controlled within the synthetic dataset generation?

**Limitations:**

yes

**Strengths And Weaknesses:**

### Strengths

- The motivation is strong. The paper cleanly identifies two key challenges in learning from implicit feedback: absence of definite negatives and heterogenous probabilities of positive actions.

- The conceptual framing is sensible and clearly motivated.

- The method of stratification is a reasonable modelling choice. The ELBO derivation provide principled grounding of the learning objective.

- The experiments are comprehensive, including multiple datasets, baselines and downstream RLHF evaluation.

### Weaknesses

1. Theorem 3.2 states unbiasedness assuming the estimated membership probabilities equal the true posteriors. The algorithm, however, learns $\hat{a}$ and $\hat{r}^*$ jointly. What is missing is perhaps an EM-style argument guaranteeing convergence to the true values

2. Purely synthetic evaluation. The implicit datasets are created from explicit datasets by sampling actions based on the ground-truth labels. The impact of noisy feedback (e.g. misclicks) is not studied.

3. Missing comparison to non-implicit feedback, as a theoretical upper bound. It would be insightful to see the gap between the proposed method learning from implicit feedback and the theoretical upper bound of learning from explicit feedback.

4. The R^ 2, MAE and RMSE metrics seem ill-suited to the preference learning setup. Ranking-focused metrics, such as the pairwise accuracy, are conventionally used for assessing reward model performance. It is also not entirely clear how the R^2, MAE and RMSE metrics are computed in this setup.

---

> ### Author Rebuttal · Authors · 2026-03-31
>
> **Thank you so much for your positive comments and for recognizing our motivation, methodology and experiments. Below are our responses to your questions.**
>
> > [W1] Missing an EM-style argument guaranteeing convergence.
>
> **Response.** We add EM-style convergence guarantee as follows. Denote $z_i := (r_i^*,a_i)\in\{\mathrm{PA,NA,PP,NP}\}$ and the stratification estimates $q_i(z_i):=\phi_i^{(z_i)}$. By Eq.10 in paper, the ELBO can be written as
> $$\mathcal{L}\_{\mathrm{ELBO}}=\frac{1}{|\mathcal{D}|}\sum_\{x_i\in\mathcal {D}}\sum\_{z_i}q_i(z_i)\,\log
> \frac{\mathbb{P}(r_i,z_i\mid x_i)}{q_i(z_i)}=\frac{1}{|\mathcal D|}\sum_{x_i\in\mathcal D}\left[\log \mathbb{P}(r_i\mid x_i)-D_{\mathrm{KL}}\bigl(q_i(z_i) \|\|\mathbb{P}(z_i\mid r_i,x_i)\bigr)\right]$$
> where the last equation applies Bayes' rule on $\mathbb{P}(r_i,z_i\mid x_i)$. Since KL divergence is nonnegative, we have $\mathcal{L}_{\mathrm{ELBO}}\leq \log \mathbb{P}(r_i\mid x_i)$
> where the RHS is log-likelihood. The equality holds if and only if $q_i(z_i)=\mathbb{P}(z_i\mid r_i,x_i)$.
>
> As training proceeds, EM iterations improve ELBO to log-likelihood, pushing $q_i(z_i)$ toward the true posterior $\mathbb{P}(r_i^*,a_i\mid r_i,x_i)$ **(E-step)**. By Theorem 3.2, this further guarantees that the learning objective converges to **an unbiased estimator of the ideal objective $\mathcal{L}_\mathrm{ideal}$ (M-step).**
>
> > [W2] Purely synthetic evaluation. The impact of noisy feedback (e.g. misclicks) is not studied.
>
> **Response.** Thank you for this helpful comment.
> - **First, we demonstrate why synthetic evaluation is necessary in this study.** Naturally implicit-feedback datasets lack test sets with both positive and negative labels, which are essential for reward-model evaluation. Nevertheless, explicit-feedback datasets provide them. We therefore synthesize implicit feedback from explicit datasets for training and use the original test sets for evaluation—a **standard practice in implicit feedback modeling [1].**
> - **Beyond the simulated setting, we add RLHF experiments** to evaluate reward models in non-simulated RL scenarios.
> - **Second, we add experiments on noisy positive feedback (e.g., misclicks).** The results (measured by $R^2$ / MAE / RMSE) on SafeRLHF yield two observations: 1) as noise ratio $\kappa$ increases, all methods degrade, yet ours remains superior to Naive. 2) Our method is compatible with denoising techniques (e.g., Co-teaching) for improved robustness against such noise.
>
> |$\kappa$|Naive|Ours|Ours+Co-teaching|
> |-|-|-|-|
> |0.02|0.525 / 0.304 / 0.343|0.687 / 0.194 / 0.278|**0.711 / 0.183 / 0.267**|
> |0.05|0.481 / 0.324 / 0.358|0.686 / 0.204 / 0.279|**0.688 / 0.203 / 0.278**|
> |0.10|0.422 / 0.359 / 0.378|0.651 / 0.232 / 0.294|**0.654 / 0.231 / 0.293**|
> |0.15|0.357 / 0.368 / 0.399|0.547 / 0.281 / 0.335|**0.609 / 0.259 / 0.311**|
>
> [1] Unbiased recommender learning from implicit feedback via weakly supervised learning. ICML.
>
> > [W3] Non-implicit feedback model.
>
> **Response.** Thank you for the comment. **We add comparison with an oracle reward model trained on non-implicit labels.** Results show that the **oracle model achieves superior performance.** Nevertheless, **ImplicitRM attains comparable performance using only implicit feedback, demonstrating its effectiveness.**
> |Metric|RMSE|R2|Accuracy|
> |-|-|-|-|
> |HelpSteer|
> |Naive|0.436|0.117|0.762|
> |ImplicitRM|0.291|0.311|0.795|
> |Oracle|0.381|0.325|0.801|
> |UltraFeedback|
> |Naive|0.423|0.145|0.781|
> |ImplicitRM|0.196|0.520|0.809|
> |Ideal|0.310|0.539|0.811|
> |SafeRLHF|
> |Naive|0.333|0.553|0.826|
> |ImplicitRM|0.105|0.787|0.857|
> |Oracle|0.214|0.814|0.861|
>
> > [W4] Pairwise accuracy, computation of $R^2$, MAE and RMSE.
>
> **Response.** Thank you for your helpful comment.
> - **First, we add experiments to evaluate pairwise accuracy.** Specifically, we follow the settings in the main table and compare ImplicitRM against Naive and four strong baselines, where ImplicitRM keeps the best pairwise accuracy.
>
> |Dataset|HelpSteer|UltraFeedback|SafeRLHF|
> |-|-|-|-|
> |Naive|0.762|0.781|0.826|
> |MTIPS|0.780|0.795|0.840|
> |SDR|0.785|0.799|0.832|
> |LaGAM|0.785|0.801|0.831|
> |SelectMix|0.788|0.802|0.844|
> |ImplicitRM|0.795|0.809|0.857|
>
> - **Second, we explain how the $R^2$, MAE and RMSE metrics are computed.** Suppose $r^\*_i$ is the true preference and $\\hat{r}_i$ is the reward model’s prediction, $\bar{r}^\*$ is the mean of $r^\*_i$.
> Given $n$ samples, the metrics are computed as:
> $$\mathrm{R}^2 = 1 - \frac{\sum\_{i=1}^n (r^*_i - \hat{r}_i)^2}{\sum\_{i=1}^n (r^\*_i - \bar{r}^\*)^2},$$
>
> $$\mathrm{MAE} = \frac{1}{n} \sum_{i=1}^n |r^*_i - \hat{r}_i|,$$
>
> $$\mathrm{RMSE} = \sqrt{\frac{1}{n} \sum_{i=1}^n (r^*_i - \hat{r}_i)^2}.$$
>
> > [Q1] How is the degree of heterogeneity controlled?
>
> **Response.**  In the training set, we sample action $a$ from $p=\alpha^{1-r^\*}$. It introduces heterogeneity in positive actions: users with larger scaled preferences ($r^\*$) have a higher probability of taking action. The degree is controlled by $\alpha$.

---

> > ### Author Rebuttal · Reviewer_upri · 2026-04-01
> >
> > The authors response have addressed my concerns.
> > That said, I believe my current rating is adequate given the scope of their contributions.

---

### Official Review · Reviewer_q25E · 2026-03-01

**Soundness:** 4
**Presentation:** 3
**Significance:** 3
**Originality:** 2
**Overall Recommendation:** 5
**Confidence:** 3

**Summary:**

The paper looks into training preference models for RLHF using implicit rather and explicit feedback. Implicit feedback means copies, likes and shares rather than explicit comparisons. There are two core challenges with implicit feedback data, no definitive negative samples and user preference bias. To solve these, they stratify each training sample into one of four latent groups based on two hidden variables, true preference (positive/negative) and action tendency (active/passive).  The authors find that this method outperforms the baselines. Ablations show that both components of the method are important for the results.

**Compliance With Llm Reviewing Policy:**

Affirmed.

**Final Justification:**

The authors mitigated most of my concerns, I think the addition of baselines and real preference data significantly strengthens the paper.

**Key Questions For Authors:**

1. Have you attempted to validate ImplicitRM on any real-world implicit feedback data, e.g. actual copy/click logs from a deployed LLM system? How confident are you that the simulation faithfully reflects real user behavior?
2. The paper assumes that observed positive feedback always indicates true positive preference. What happens if we assume this signal is noisy?
3. How does ImplicitRM compare against a reward model trained on the full explicit preference labels?
4. Direct alignment methods like DPO and KTO bypass reward modeling entirely. Could you compare this to ImplicitRM?

**Limitations:**

yes

**Strengths And Weaknesses:**

Strengths:
- Important and timely research topic
- Comprehensive experiments and ablations

Weaknesses:
- Simulated implicit feedback. The implicit preference scenario appears to be entirely simulated. Real user behavior would likely be far messier bringing into question whether the method would work with a real world deployment.
- The paper could do with more comparisons to baselines/other alignment methods.
- Eq. 9 strikes me as an odd assumption. I don't think action and preference should be seen as independent given the input.

---

> ### Author Rebuttal · Authors · 2026-03-30
>
> **Thank you so much for your positive comments and for recognizing our significance and experiments. Below are our responses to your concerns and questions.**
>
> ---
>
> > [W1,Q1] Simulated implicit feedback. How confident are you that the simulation reflects real user behavior?
>
> **Response.** Thank you very much for this valuable comment.
> - **First, we demonstrate why simulated implicit datasets are necessary for our experiments.** The naturally implicit-feedback datasets lack test sets with explicit positive and negative labels, which are indispensable for model evaluation. Therefore, we have to synthesize implicit feedback from explicit-feedback datasets for training while retaining the original test sets for evaluation.
> - **Second, we believe this protocol reflects user behaviors for three reasons.** 1) It preserves all real-world prompts and LLM responses, only masking labels to generate implicit feedback. 2) The simulation follows the standard process in implicit feedback modeling research [1]. 3) Beyond the simulated setting, we add RLHF experiments, directly verifying the reward model's effectiveness in real-world RL scenarios.
>
> [1] Unbiased recommender learning from implicit feedback via weakly supervised learning. ICML.
>
> > [W2] More comparisons to baselines.
>
> **Response.** Thank you for your actionable comment. **We add experiments on three recent baselines.** The results below (measured by $R^2$ / MAE / RMSE) show that ImplicitRM retains superior performance over them.
>
> |Method|HelpSteer|SafeRLHF|UltraFeedback|
> |-|-|-|-|
> |OME-IPS [2]|0.248 / 0.339 / 0.403|0.726 / 0.138 / 0.261|0.358 / 0.303 / 0.367|
> |OME-DR [2]|0.247 / 0.355 / 0.403|0.704 / 0.168 / 0.271|0.351 / 0.247 / 0.369|
> |Labelwave [3]|0.271 / 0.345 / 0.397|0.746 / 0.136 / 0.251|0.466 / 0.233 / 0.335|
> |ImplicitRM|**0.311 / 0.292 / 0.386**|**0.787 / 0.105 / 0.229**|**0.521 / 0.196 / 0.317**|
>
> [2] Debiased recommendation with noisy feedback. KDD. 2024.
>
> [3] Early stopping against label noise without validation data. ICLR. 2025.
>
> > [W3] Eq. 9: I do not think action ($a$) and preference ($r^*$) are independent given $x$.
>
> **Response.** Thank you for your meticulous comment. We clarify below why the conditional independence holds. The probabilistic graphical model for the data-generating process is
> $$a \leftarrow x \rightarrow r^*$$
>
> where both $a$ and $r^\star$ depend on $x$; while the action $a$ does not directly impacts preference $r^\star$.
>  This is a standard `fork` structure, which implies $a$ and $r^*$ are independent conditioned on $x$.
>
> > [Q2] What if positive feedback is noisy?
>
> **Response.** **We add experiments to evaluate model performance under noisy positive feedback.** The results (measured by $R^2$ / MAE / RMSE) yield two observations: 1) as noise ratio $\kappa$ increases, the performance of all methods degrades; however, our method remains superior to Naive. 2) Our method can be combined with denoising techniques (e.g., Co-teaching) to improve robustness against such noise.
>
> |$\kappa$|Naive|Ours|Ours+Co-teaching|
> |-|-|-|-|
> |SafeRLHF|
> |0.02|0.525 / 0.304 / 0.343|0.687 / 0.194 / 0.278|**0.711 / 0.183 / 0.267**|
> |0.05|0.481 / 0.324 / 0.358|0.686 / 0.204 / 0.279|**0.688 / 0.203 / 0.278**|
> |0.10|0.422 / 0.359 / 0.378|0.651 / 0.232 / 0.294|**0.654 / 0.231 / 0.293**|
> |0.15|0.357 / 0.368 / 0.399|0.547 / 0.281 / 0.335|**0.609 / 0.259 / 0.311**|
> |HelpSteer|
> |0.02|0.112 / 0.405 / 0.437|0.304 / 0.295 / 0.387|**0.319 / 0.289 / 0.383**|
> |0.05|0.096 / 0.415 / 0.441|0.299 / 0.296 / 0.389|**0.312 / 0.292 / 0.385**|
> |0.10|0.057 / 0.430 / 0.451|0.297 / 0.296 / 0.389|**0.311 / 0.292 / 0.385**|
> |0.15|0.011 / 0.448 / 0.462|0.289 / 0.300 / 0.391|**0.306 / 0.294 / 0.387**|
>
> > [Q3] Compare with training on explicit labels.
>
> **Response.** **We add comparison with an Oracle model trained on explicit labels.** Although the oracle achieves superior performance, ImplicitRM attains comparable results using only implicit feedback, which further demonstrates its effectiveness.
> |Metric|RMSE|MAE|R2|Accuracy|
> |-|-|-|-|-|
> |**HelpSteer**|
> |Naive|0.4366|0.3907|0.1179|0.7628|
> |ImplicitRM|0.3857|0.2919|0.3114|0.7959|
> |Oracle|0.3819|0.2885|0.3250|0.8010|
> |**UltraFeedback**|
> |Naive|0.4230|0.3492|0.1459|0.7813|
> |ImplicitRM|0.3169|0.1961|0.5207|0.8090|
> |Ideal|0.3105|0.1889|0.5399|0.8119|
> |**SafeRLHF**|
> |Naive|0.3330|0.2887|0.5535|0.8263|
> |ImplicitRM|0.2294|0.1053|0.7872|0.8570|
> |Oracle|0.2145|0.0933|0.8146|0.8618|
>
>
> > [Q4] Compare with DPO and KTO.
>
> **Response.** Thank you for your helpful comment. **We add experiments comparing with DPO and KTO.** Since DPO relies on pairwise labels that are inavailable in our setting, we apply DPO and KTO on explicit feedbacks. The results show that, despite not using explicit feedback, ImplicitRM still outperforms them, demonstrating the advantage of RL.
> |Benchmark|HarmBench|StrongReject|WildGuardMix|
> |-|-|-|-|
> |Naive|83.81%|90.07%|76.54%|97.00%|
> |ImplicitRM|92.58%|97.10%|88.27%|
> |DPO|90.78%|90.22%|70.97%|
> |KTO|93.33%|94.64%|74.19%|

---

> > ### Author Rebuttal · Reviewer_q25E · 2026-04-02
> >
> > The authors have addressed some of my concerns, the addition of baselines is great.
> > wrt eq 9:
> > I wanted to see a more careful discussion of whether the DAG itself is justified, not just that the DAG implies conditional independence. The question isn't really about d-separation mechanics; it's about whether the assumed causal structure is realistic. Do action propensities truly have no direct effect on preference, or vice versa?
> > wrt w1:
> > Reason 1 doesn't actually validate that the simulation reflects real user behavior, it just says the content is real while the feedback mechanism is synthetic. Reason 3: The RLHF experiments show the learned reward model is useful for downstream RL, but they don't directly validate that the simulated implicit feedback mirrors real user behavior.

---

> > > ### Author Response · Authors · 2026-04-02
> > >
> > > Dear reviewer q25E,
> > >
> > > Thank you for your valuable suggestion to improve the assumption justification and real-world evaluation. **We have carefully acted on your feedback, expanded discussion on DAG assumption, and added experiments on our private real-world production data.** The detailed response  is presented below.
> > >
> > >
> > > > [Q1] I wanted to see a more careful discussion of whether the DAG itself is justified, not just that the DAG implies conditional independence.
> > >
> > >
> > > **Response.**  Thank you for this important clarification. We agree that it is essential to justify the DAG itself, rather than only the conditional independence it implies. In our formulation, both $a$ and $r^\*$ depend on $x$, i.e., the user action and the latent true preference depend on the prompt–response pair. Here, $r^\*$ denotes the underlying true user preference here, rather than the observed feedback $r$. The remaining assumption is that there is no direct edge from $a$ to $r^\*$. Notably,
> > >
> > > - **First, we would clarify that this assumption becomes plausible when $x$ contains all causes of $a$ and $r^\*$**. Under this condition, once $x$ is given, knowing $a$ provides no additional information about $r^*$, since all factors affecting $r^\*$ have already been captured by $x$, which renders $a$ and $r^\*$ independent.
> > > - One could posit that it may be difficult to guarantee that $x$ contains all such causes in practice. **Nevertheless, this assumption is standard in the implicit feedback modeling literature [1–5]**, where it plays a role analogous to the unconfoundedness assumption in causal inference. Specifically, prior work commonly models the observed feedback as:
> > > $P(r=1 \mid x) = P(a=1 \mid x) \cdot P(r^\*=1 \mid x),$
> > > **which directly assumes conditional independence between $a$  and $r^\*$ given $x$.** This formulation is adopted, for example, in Eq. 2 of [1–3], Eq. 1 of [4], and Eq. 6 of [5].
> > > As a side note, the notation in [1-5] is written per sample as $P(r_{u,i}=1) = P(a_{u,i}=1) \cdot P(r^\*_{u,i}=1)$; in our paper, the sample-specific information is represented by the covariate $x$, so this formulation is equivalent to conditioning on $x$.
> > >
> > > - To further improve rigor and clarity, we will revise Theorem 3.1 to explicitly state this assumption, clearly delineate the scope of the theoretical analysis, and further emphasize it in the discussion of limitations.
> > >
> > >
> > >
> > >
> > > [1] Unbiased Pairwise Learning from Biased Implicit Feedback. SIGIR 2020.
> > >
> > > [2] Bilateral Self-unbiased Learning from Biased Implicit Feedback. SIGIR 2022.
> > >
> > > [3] Unbiased Recommender Learning from Missing-Not-At-Random Implicit Feedback. WSDM 2020.
> > >
> > > [4] Unbiased Pairwise Learning from Implicit Feedback for Recommender Systems without Biased Variance Control. SIGIR 2023.
> > >
> > > [5] Dual Unbiased Recommender Learning for Implicit Feedback. SIGIR 2021.
> > >
> > >
> > >
> > > > [W1] Simulated implicit feedback. The implicit preference scenario appears to be entirely simulated.
> > >
> > > **Response.** Thank you very much for your valuable comment - we truly appreciate it. To fully address this concern, **we add experiments on real-world industrial data.** Specifically, we appeal for the implicit preference dataset collected from our online industrial LLM service as the training set, while an evaluation set curated by its product team was used as the test set. The table below compares ImplicitRM with Naive and four strong baselines on this industrial dataset. **The results show that ImplicitRM maintains superior performance, validating its effectiveness on real-world implicit feedback data.**
> > >
> > > |  | $R^2$ | MAE | RMSE | Pairwise Accuracy |
> > > | --- | --- | --- | --- | --- |
> > > | Naive | 0.1220 | 0.3647 | 0.4206 | 0.7210 |
> > > | MTIPS | 0.2548 | 0.3576 | 0.3875 | 0.7390 |
> > > | SDR2 | 0.2725 | 0.3432 | 0.3829 | 0.7410 |
> > > | LaGAM | 0.4646 | 0.2427 | 0.3285 | 0.7495 |
> > > | ILDE | 0.4585 | 0.2198 | 0.3303 | 0.7470 |
> > > | ImplicitRM | **0.5011** | **0.2021** | **0.3170** | **0.7535** |
> > >
> > >
> > > ---
> > >
> > > We sincerely appreciate and value your initial positive and detailed comments. Your suggestions on real-world data evaluation were especially helpful, which makes this work much more comprehensive as a result. If our response and additional experiments have properly resolved your raised issues, we politely invite you to further increase your support - thank you!
> > >
> > > Submission 34165 Authors

---

### Official Review · Reviewer_Vid4 · 2026-03-10

**Soundness:** 3
**Presentation:** 3
**Significance:** 2
**Originality:** 3
**Overall Recommendation:** 4
**Confidence:** 3

**Summary:**

The paper studies reward modeling for reinforcement learning from human feedback using implicit user feedback instead of explicit preference labels. Implicit feedback, such as clicks or copy actions, is cheaper to collect but introduces two main challenges: the absence of clear negative samples and bias caused by different probabilities of users giving feedback.
To solve these issues, the authors propose ImplicitRM, a method that stratifies samples into four latent groups based on user preference and action behavior and learns reward models through a likelihood-based objective. Experiments show that this approach produces more accurate reward models and improves downstream RLHF performance compared with existing methods.

**Compliance With Llm Reviewing Policy:**

Affirmed.

**Final Justification:**

The paper investigates reward modeling for reinforcement learning from human feedback (RLHF) using implicit user signals rather than explicit preference annotations. While implicit feedback (e.g., clicks or copy actions) is more scalable to collect, it introduces two key challenges: the lack of explicit negative samples and biases arising from heterogeneous user feedback propensities.

To address these issues, the authors propose ImplicitRM, which stratifies data into four latent groups based on user preference and interaction behavior, and optimizes a likelihood-based objective to learn the reward model. Experimental results demonstrate that this method yields more accurate reward estimates and improves downstream RLHF performance compared to existing approaches.

A primary concern is that much of the framework and experimentation is conducted in a simulated sandbox environment, which may oversimplify real-world conditions. However, the authors effectively mitigate this concern by further validating their approach on a real-world industry test set, providing stronger evidence of practical applicability.

**Key Questions For Authors:**

Please refer to the “Strengths and Weaknesses” section. I would be willing to reconsider and potentially increase the rating if the authors adequately address the concerns I raised.

**Limitations:**

yes

**Strengths And Weaknesses:**

Strength
1. The paper clearly explains the problem and motivates the method well, with a logical structure that makes the approach easy to follow.
2. The experiments are comprehensive, including multiple baselines, datasets, and ablation studies to validate the effectiveness of the proposed method.

Weakness
1. The definition of feedback is not sufficiently clear. The paper states that implicit preference signals (e.g., clicks or copy actions) consist only of positive feedback and no-feedback, but it does not provide a rigorous definition of these signals or the complete set of user actions used in the experiments. In addition, prior work in search and recommendation systems often interprets user behaviors such as clicks as both positive and negative signals through appropriate polarization strategies [1]. This weakens the argument that the absence of explicit negative feedback constitutes a unique challenge.
2. The paper does not clearly explain how implicit reward modeling reduces data collection costs. If preference data are obtained through user feedback, it is unclear how this setting fundamentally differs from traditional ranking-based approaches, where the same feedback signals are commonly used to construct pairwise comparison data.
3. The paper evaluates reward prediction using RMSE, MAE, and R^2 to measure consistency with binary ground-truth labels, but reward models in RLHF are commonly used for comparison or ranking. The main experiments therefore lack more direct reward-model metrics such as pairwise preference accuracy or win rate. Reporting preference win rate would provide a more relevant evaluation [2].
4. The unbiasedness claim relies on the assumption that the estimated stratification probabilities match the true posterior, yet the paper does not analyze whether this posterior is identifiable or guaranteed to converge. Additional discussion on the uniqueness of the posterior and on whether the stop-gradient approximation could cause the ELBO optimization to deviate from the true optimum would strengthen the theoretical justification.

[1] Guo, X., Chen, B., Wang, S., Yang, Y., Lei, C., Ding, Y., & Li, H. (2025). Onesug: The unified end-to-end generative framework for e-commerce query suggestion. arXiv preprint arXiv:2506.06913.
[2] Lambert, N., Pyatkin, V., Morrison, J., Miranda, L. J. V., Lin, B. Y., Chandu, K., ... & Hajishirzi, H. (2025, April). Rewardbench: Evaluating reward models for language modeling. In Findings of the Association for Computational Linguistics: NAACL 2025 (pp. 1755-1797).

TYPO
1. There appears to be a typo in the formulation of PP in Eq. (4).

---

> ### Author Rebuttal · Authors · 2026-03-30
>
> **Thank you very much for your meticulous review and for recognizing our motivations, logical structure, and experiments. Below are our responses to your concerns and queries.**
>
> ---
>
> > [W1] The paper lacks definition of implicit feedback signals. Besides, prior polarization strategies would weaken challenge.
>
> **Response.** Thank you very much for your helpful comment.
> - **First, we clarify the definition of implicit feedback signals.** Users have two possible actions: taking a positive action or taking no action. Accordingly, the feedback $r$ is binary: $r=1$ if positive feedback is observed, and $r=0$ otherwise. Notably, $r=0$ does not imply negative user preference, as a user may still hold a positive preference without taking any action. In our experiments, implicit feedback is simulated by assigning each instance an action propensity ($a$). An instance is recorded as positive feedback ($r=1$) only when the user takes an action ($a=1$) and the true preference is positive (e.g., rating is greater than 4). All remaining instances are recorded as no-feedback ($r=0$).
> - **One may ask why we do not use naturally implicit-feedback datasets for experiments.** The reason is that such datasets lack test sets with explicit positive and negative labels, which are essential for model evaluation. Therefore, we synthesize implicit feedback from explicit-feedback datasets for training while using the original test sets for evaluation. This protocol aligns with standard practice in implicit feedback modeling [1].
> - **Second, we clarify why polarization strategies do not mitigate the challenge considered in ImplicitRM.** The strategies like OneSug rely on **intermediate pipeline labels** (e.g., recall or exposure) to polarize negative labels. Such signals are unavailable in our reward modeling setting, which does not include recall or ranking stages. **Therefore, the absence of negative feedback remains an unresolved challenge which cannot be addressed by polarization strategies.**
>
> [1] Unbiased recommender learning from implicit feedback via weakly supervised learning. ICML.
>
> > [W2] Reduction of data collection costs; difference from prior works based on pairwise data.
>
> **Response.** Thank you very much for your careful comment.
> - **First, we explain how our setting reduces data collection costs.** As you noted, most ranking-based methods rely on **pairwise comparison data**, which is expensive to collect because it interrupts users’ natural interactions with LLMs and requires them to read and compare two responses. In contrast, our study relies only on **(positive) pointwise feedback**, which can be collected directly from users’ natural interactions without interruption. **Such feedback is less intrusive to obtain and thus reducing collection cost.**
> - **Second, we summarize difference from prior works.** Prior works mostly rely on pairwise comparison data, whereas our method uses only pointwise (positive) feedback.
>
>
>
> > [W3] Metrics.
>
> **Response.** **We add experiments to evaluate pairwise accuracy.** Specifically, we follow the settings in the main table and compare ImplicitRM against Naive and four strong baselines, where ImplicitRM retains the best pairwise accuracy. We promise to add these results in the revised manuscript.
>
> |Dataset|HelpSteer|UltraFeedback|SafeRLHF|
> |-|-|-|-|
> |Naive|0.7628|0.7813|0.8263|
> |MTIPS|0.7806|0.7957|0.8406|
> |SDR|0.7857|0.7998|0.8326|
> |LaGAM|0.7857|0.8015|0.8316|
> |SelectMix|0.7883|0.8027|0.8443|
> |ImplicitRM|0.7959|0.8090|0.8570|
>
> > [W4] Stratification probability convergence, uniqueness and stop-gradient approximation.
>
> **Response.** Thank you for your valuable comment. Denote $z_i := (r_i^*,a_i)\in\{\mathrm{PA,NA,PP,NP}\}$ and the stratification estimates $q_i(z_i):=\phi_i^{(z_i)}$. By Eq.10 in paper, the ELBO can be written as
> $$\mathcal{L}\_{\mathrm{ELBO}}=\frac{1}{|\mathcal{D}|}\sum_\{x_i\in\mathcal {D}}\sum\_{z_i}q_i(z_i)\,\log
> \frac{\mathbb{P}(r_i,z_i\mid x_i)}{q_i(z_i)}=\frac{1}{|\mathcal D|}\sum_{x_i\in\mathcal D}\left[\log \mathbb{P}(r_i\mid x_i)-D_{\mathrm{KL}}\bigl(q_i(z_i) \|\| \mathbb{P}(z_i\mid r_i,x_i)\bigr)\right]$$
> where the last equation applies Bayes' rule: $\mathbb{P}(r_i,z_i\mid x_i)=\mathbb{P}(z_i\mid r_i,x_i)\mathbb{P}(r_i\mid x_i)$.
>
> Since KL divergence is nonnegative, we have $\mathcal{L}_{\mathrm{ELBO}}\leq \log \mathbb{P}(r_i\mid x_i)$, where the RHS is log-likelihood. The equality holds if and only if $q_i(z_i)=\mathbb{P}(z_i\mid r_i,x_i)$.
> As training proceeds, the EM iterations improve ELBO to log-likelihood, so the stratification estimates are pushed toward the true posterior $\mathbb{P}(r_i^*,a_i\mid r_i,x_i)$, **which proves uniqueness and convergence of stratification probability**. **Stop-gradient is used to prevent the M-step updates from influencing the E-step, which is a common practice in EM and essential for convergence.**
>
>
> > [W5] A typo in Eq.4.
>
> **Response.** Thank you so much for pointing them out. We will immediately correct it in revision.

---

> > ### Author Rebuttal · Reviewer_Vid4 · 2026-04-01
> >
> > Thank you for the clarification. Most of my concerns have been addressed, but one issue remains only partially resolved. The framework and experiments are conducted in a simulated sandbox environment, which appears overly simplified and may not fully reflect real-world complexity.

---

> > > ### Author Response · Authors · 2026-04-02
> > >
> > > Dear reviewer Vid4,
> > >
> > > Thank you for your swift follow-up feedback. **We are delighted that only one concern remains partially resolved:**
> > >
> > > > **The framework and experiments are conducted in a simulated sandbox environment, which appears overly simplified and may not fully reflect real-world complexity.**
> > >
> > > **To fully address this concern, we add experiments on real-world industrial data.** Specifically, we appeal for the implicit preference dataset collected from our online industrial LLM service as the training set, while an evaluation set curated by its product team was used as the test set. The table below compares ImplicitRM with Naive and four strong baselines on this industrial dataset. **The results show that ImplicitRM maintains superior performance, validating its effectiveness on real-world implicit feedback data.**
> > >
> > > |  | R2 | MAE | RMSE | Pairwise Accuracy |
> > > | --- | --- | --- | --- | --- |
> > > | Naive | 0.1220 | 0.3647 | 0.4206 | 0.7210 |
> > > | MTIPS | 0.2548 | 0.3576 | 0.3875 | 0.7390 |
> > > | SDR2 | 0.2725 | 0.3432 | 0.3829 | 0.7410 |
> > > | LaGAM | 0.4646 | 0.2427 | 0.3285 | 0.7495 |
> > > | ILDE | 0.4585 | 0.2198 | 0.3303 | 0.7470 |
> > > | ImplicitRM | **0.5011** | **0.2021** | **0.3170** | **0.7535** |
> > >
> > > ---
> > >
> > > We sincerely appreciate your initial encouragement and insightful comments. Your suggestions on real-world data evaluation, explicit feedback baseline, and theoretical justification were especially helpful, which makes this work much more rigorous and comprehensive as a result. If our response and additional experiments have properly resolved your raised issues, we politely invite you to further reconsider your overall rating - thank you!
> > >
> > > Submission 34165 Authors

---

### Official Review · Reviewer_Fcgw · 2026-03-13

**Soundness:** 2
**Presentation:** 2
**Significance:** 3
**Originality:** 3
**Overall Recommendation:** 4
**Confidence:** 3

**Summary:**

To address the high cost of obtaining explicit human preference data for large language model alignment, this work explores leveraging abundant, low-cost implicit feedback (e.g., copying, clicking) for training reward models. It identifies and tackles two fundamental challenges of implicit data—"the absence of definitive negative samples" and "significant user feedback bias (action propensity heterogeneity)"—by proposing the ImplicitRM​ framework for unbiased reward modeling. The method introduces latent variables to stratify samples into four groups based on true preference and user action propensity. Using variational inference, it derives an Evidence Lower Bound (ELBO) for the log marginal likelihood as the optimization objective, which alternates training between a preference estimator and an action propensity estimator. The approach demonstrates significantly higher reward prediction accuracy than various debiasing and Positive-Unlabeled (PU) learning baselines on multiple synthetic implicit datasets. Furthermore, it guides downstream RLHF policy fine-tuning to achieve superior safety alignment performance.

**Compliance With Llm Reviewing Policy:**

Affirmed.

**Final Justification:**

This paper proposes the ImplicitRM framework for training reward models from noisy implicit human feedback. It addresses the joint challenge of false negatives (PU learning) and user preference heterogeneity (selection bias) by stratifying samples with latent variables and deriving an ELBO objective to debias reward estimation.

The rebuttal thoroughly resolved my concerns. I initially rated the paper as a reject due to its reliance on synthetic data and the parameterization of observationally indistinguishable latent states. The authors justified their modeling choices by analogy to zero-inflated Poisson models, corrected my misinterpretation of the sensitivity analysis, and provided validation on a real-world industrial dataset, thus addressing the "synthetic data limitation" concern.

As all initial issues are resolved, I raise my rating from 2 (Reject) to 4 (Weak accept).

**Key Questions For Authors:**

See the "Weaknesses".

**Limitations:**

yes

**Strengths And Weaknesses:**

**Strengths**:
1.  The problem of implicit feedback is insightfully reframed as a superposition of the "false negative (PU Learning) problem" and "user propensity heterogeneity (Selection Bias) problem."
2.  A solid theoretical framework is established by deriving the ELBO through latent variable stratification, supported by the unbiasedness proof in Theorem 3.2.
3.  Extensive experimental comparisons with over ten traditional debiasing methods (IPS, DR, etc.) and PU learning methods (BPR, UPU, nnPU, etc.).

**Weaknesses**:
1.  The paper claims to address the "implicit feedback" problem but does not employ any real-world implicit interaction dataset. Consequently, the strong performance of ImplicitRM lacks persuasiveness. Validation on real user chat logs is needed, or at minimum, the propensity distribution for synthetic data generation should be explicitly stated, and the model's robustness under distribution shift should be tested.
2.  Conceptual confusion surrounds the "Negative & Active (NA)" group. "Refusing to copy" resembles a mental state rather than an observable system action. In a binary implicit feedback system (only "copy=1" and "no action=0"), "wanting to copy but refraining (NA)" and "skipping without reading (NP)" are physically indistinguishable observations. Treating NA and NP as independent categories in the ELBO derivation constitutes physically meaningless over-parameterization without additional distinguishing features.
3.  Figure 2(a) reveals that ImplicitRM exhibits high sensitivity to the update rate η. This fragile convergence property implies the method would be unstable and difficult to tune for a new, unseen implicit dataset.

---

> ### Author Rebuttal · Authors · 2026-03-30
>
> **Thank you very much for your thoughtful review and for recognizing our insights, theoretical analysis and experiments. Below are our responses to your specific concerns.**
>
> ---
>
> > [W1] **Validation on real user chat logs** is needed, or at minimum, **the propensity distribution for synthetic data generation** should be explicitly stated, and **the model's robustness under distribution shift** should be tested.
>
> **Response.** Thank you very much for your valuable comment.
> - **Firstly, we would clarify that validation is performed on real-world chatlog datasets.** **Implicit feedback is synthesized only for training:** the original real-world prompts and LLM responses are preserved, while only the explicit labels are converted into implicit ones. **This design is both necessary and consistent with standard practice in implicit feedback modeling.** For example, [1] generates implicit feedback from explicit feedback datasets (e.g., Coat and KuaiRec) to construct training data. This approach is necessary because **it preserves test sets with explicit positive and negative labels, which are essential for evaluating reward model performance. In contrast, evaluation on purely implicit feedback datasets is infeasible because explicit negative labels are missing.**
> - **Secondly, we state the propensity distribution for synthetic data generation.**  In the training set, the propensity—defined as the probability that a user takes an action—is generated as $p=\alpha^{1-r*}$, where $r*$ is the scaled ground-truth user rating or preference for the LLM response. Under this formulation, propensity varies across users: those with stronger preferences have a higher probability of taking action.
> - **Thirdly, we add experiments to test model robustness under distribution shift.**  Specifically, as $\alpha$ decreases, the propensity distribution deviates further from the uniform distribution, thereby increasing the distribution shift between the training and test sets. The results below (measured by $R^2$) show that as $\alpha$ decreases from 0.9 to 0.1, ImplicitRM consistently outperforms Naive and exhibits substantially smaller performance fluctuations, demonstrating greater robustness to distribution shift.
>
>
> | $\alpha$|0.1 |0.3|0.5|0.7| 0.9 |
> |-|-|-|-|-|-|
> |UFB data|
> |Naive|-0.1982|-0.1963|0.1627|0.3524|0.4411 |
> |ImplicitRM|**0.4697**|**0.4896**|**0.5003**|**0.5232**|**0.5281** |
> |SafeRLHF data |
> |Naive|0.0434|0.2262|0.3144|0.5877|0.6895 |
> |ImplicitRM|**0.6980**|**0.7186**|**0.7430**|**0.7992**|**0.8065** |
>
> [1] Unbiased recommender learning from implicit feedback via weakly supervised learning. ICML.
>
> > [W2] Treating NA and NP as independent categories constitutes **meaningless over-parameterization.**
>
> **Response.** Thank you very much for your meticulous comment!
> - **Firstly, we posit that independently parameterizing NA and NP probabilities is necessary.** In the loss function below (Eq.7), $\phi^\mathrm{(NP)}_i$ and $\phi^\mathrm{(NA)}_i$ denote the membership probabilities of NP and NA, respectively. Without modeling them independently, the loss function cannot be computed. **Therefore, treating NA and NP as independent variables is not a meaningless over-parameterization, but a necessary design choice.**
>
> $\mathcal{E}\_\mathrm{prop}(\psi)=-\frac{1}{|\mathcal{D}|} \sum\_{x_i\in\mathcal{D}} (\phi^\mathrm{(PA)}_i+\phi^\mathrm{(NA)}_i)\log\hat{a}\_\psi(x_i)-\frac{1}{|\mathcal{D}|} \sum\_{x_i\in\mathcal{D}}+(\phi^\mathrm{(PP)}_i+\phi^\mathrm{(NP)}_i)\log(1-\hat{a}\_\psi(x_i))$.
>
> - **Secondly, we clarify that parameterizing latent states that are observationally indistinguishable is a standard practice in latent variable modeling.** A classical example is the zero-inflated Poisson model, where an observed zero may arise either from a structural-zero latent state or from the Poisson-count component; although these two causes are observationally identical, they are modeled separately because they correspond to different data-generating mechanisms, and inference proceeds by estimating their posterior responsibilities.
>
>
> > [W3] Figure 2 reveals that **ImplicitRM is sensitve to $\eta$.** It would be unstable and difficult to tune for a new dataset.
>
> **Response.** Thank you for this comment. It seems **Figure 2 suggests that ImplicitRM is insensitve to $\eta$,** rather than sensitive to it. As $\eta$ increasesfrom 0.005 to 0.5, the $R^2$ score decreases only slightly, from 0.30 to 0.27, indicating substantially greater stability than baselines such as Naive. **Furthermore, we add varying $\eta$ experiments on UFB dataset below**, where ImplicitRM remains stable $R^2$ to $\eta$ variation. These findings suggest that the model is robust to $\eta$ and can be tuned easily on new datasets.
>
> |$\eta$|0.005|0.01|0.05|0.1|0.5|
> |-|-|-|-|-|-|
> |HelpSteer|
> |Naive|0.029|0.055|0.110|0.146|0.222 |
> |ImplicitRM|0.308|0.289|0.267|0.300|0.274|
> |UltraFeedback|
> |Naive|0.167|0.176|0.238|0.285|0.284 |
> |ImplicitRM|0.498|0.508|0.521|0.483|0.501|

---

> > ### Author Rebuttal · Reviewer_Fcgw · 2026-04-03
> >
> > Thank you for your comprehensive and convincing rebuttal. The additional clarifications and data you provided have fully addressed my concerns and corrected my initial misapprehension. My apologies for the earlier mistake in judgment. I will therefore update my score to reflect this positive change.

---

> > > ### Author Response · Authors · 2026-04-03
> > >
> > > Dear reviewer Fcgw,
> > >
> > > Thank you very much for your thoughtful and swift follow-up feedback.  **We are delighted that our response effectively addresses your concerns.**
> > >
> > > **We would also like to share an additional result to further address the concern raised in [W1]: we have conducted experiments on real-world industrial data.** Specifically, we secured access to an implicit preference dataset collected from our online industrial LLM service and used it as the training set, while an evaluation set curated by the product team was used as the test set. The table below compares ImplicitRM with Naive and four strong baselines on this industrial dataset. **The results show that ImplicitRM maintains superior performance, validating its effectiveness on real-world implicit feedback data.**
> > >
> > > |  | R2 | MAE | RMSE | Pairwise Accuracy |
> > > | --- | --- | --- | --- | --- |
> > > | Naive | 0.1220 | 0.3647 | 0.4206 | 0.7210 |
> > > | MTIPS | 0.2548 | 0.3576 | 0.3875 | 0.7390 |
> > > | SDR2 | 0.2725 | 0.3432 | 0.3829 | 0.7410 |
> > > | LaGAM | 0.4646 | 0.2427 | 0.3285 | 0.7495 |
> > > | ILDE | 0.4585 | 0.2198 | 0.3303 | 0.7470 |
> > > | ImplicitRM | **0.5011** | **0.2021** | **0.3170** | **0.7535** |
> > >
> > > ---
> > >
> > >
> > > We sincerely appreciate your actionable and insightful comments. In particular, your suggestions on real-world data evaluation, simulation protocol clarification, and sensitivity analysis were especially helpful, which makes this work much more detailed and comprehensive as a result. If our recently added experiments have further resolved your raised issues, we politely invite you to further improve your support - thank you!
> > >
> > > Submission 34165 Authors

---

### Decision · Program_Chairs · 2026-04-30

**Decision:**

Accept (regular)

**Comment:**

This paper studies reward modeling from implicit feedback rather than explicit preference labels and proposes ImplicitRM, a principled framework that stratifies samples into latent groups and derives an unbiased learning objective; the reviewers found the problem timely, the formulation well motivated, and the experiments comprehensive, with clear gains on reward modeling and downstream RLHF. The main concerns were the reliance on simulated implicit data, assumptions in the latent-variable formulation, and the choice of evaluation metrics, but the rebuttal substantially addressed these points with additional clarification, stronger baselines, and real-world industrial data validation, leaving limitations that are appropriate for discussion in the final version.